# Hypothermia: Beyond the Narrative Review—The Point of View of Emergency Physicians and Medico-Legal Considerations

**DOI:** 10.3390/jpm13121690

**Published:** 2023-12-05

**Authors:** Gabriele Savioli, Iride Francesca Ceresa, Gaia Bavestrello Piccini, Nicole Gri, Alba Nardone, Raffaele La Russa, Angela Saviano, Andrea Piccioni, Giovanni Ricevuti, Ciro Esposito

**Affiliations:** 1Emergency Department, IRCCS Policlinico San Matteo, 27100 Pavia, Italy; 2Emergency Department and Internal Medicine, Istituti Clinici di Pavia e Vigevano, Gruppo San Donato, 27029 Vigevano, Italy; irideceresa@gmail.com; 3Emergency Medicine, Université Libre de Bruxelles, 1050 Brussels, Belgium; 4Niguarda Cancer Center, ASST Grande Ospedale Metropolitano Niguarda, Piazza dell’Ospedale Maggiore, 3, 20162 Milano, Italy; 5Emergency Department, Ospedale Civile, 27058 Voghera, Italy; 6Department of Clinical and Experimental Medicine, Section of Forensic Pathology, University of Foggia, 71122 Foggia, Italy; 7Emergency Department, Fondazione Policlinico Universitario A. Gemelli, IRCCS, 00168 Roma, Italy; angela.saviano@policlinicogemelli.it (A.S.); andrea.piccioni@policlinicogemelli.it (A.P.); 8Department of Drug Science, University of Pavia, 27100 Pavia, Italy; giovanni.ricevuti@unipv.it; 9Nephrology and Dialysis Unit, ICS Maugeri, University of Pavia, 27100 Pavia, Italy; ciro.esposito@unipv.it

**Keywords:** hypothermia, emergency department

## Abstract

Hypothermia is a widespread condition all over the world, with a high risk of mortality in pre-hospital and in-hospital settings when it is not promptly and adequately treated. In this review, we aim to describe the main specificities of the diagnosis and treatment of hypothermia through consideration of the physiological changes that occur in hypothermic patients. Hypothermia can occur due to unfavorable environmental conditions as well as internal causes, such as pathological states that result in reduced heat production, increased heat loss or ineffectiveness of the thermal regulation system. The consequences of hypothermia affect several systems in the body—the cardiovascular system, the central and peripheral nervous systems, the respiratory system, the endocrine system and the gastrointestinal system—but also kidney function, electrolyte balance and coagulation. Once hypothermia is recognized, prompt treatment, focused on restoring body temperature and supporting vital functions, is fundamental in order to avert preventable death. It is important to also denote the fact that CPR has specificities related to the unique profile of hypothermic patients.

## 1. Introduction

Hypothermia is a widespread condition observed all over the world, with a high risk of mortality in pre-hospital and in-hospital settings when its treatment is not implemented quickly and adequately [1]. It has been estimated that the incidence of mortality in hypothermic patients fluctuates between 13.3% and 43% in various prehospital settings around the world [2,3,4,5]. In a prehospital setting, the risk for hypothermia increases exponentially in the case of disease as well as of trauma, especially major trauma [6,7,8].

Hypothermia affects more people in the United States of America than complications from exposure to heat; it is estimated that hypothermia causes twice as many deaths as heatstroke (over 1300 deaths per year) [9].

In New Zealand, 0.13–6.9 cases of hypothermia per 100,000 inhabitants per year are observed [10].

In the United Kingdom, the estimated number of deaths attributable to hypothermia is about 1.81–2.2 per 100,000 inhabitants per year [11].

Hypothermia also affects other European countries with high prevalence; for example, Poland has 0.88–1.6 deaths per 100,000 inhabitants a year [12].

The aim of this review is to describe the main specificities of the diagnosis and treatment of hypothermia through consideration of the physiological changes that occur in hypothermic patients.

## 2. Materials and Methods

Our review is based on an analysis of publications on the topic of hypothermia through the principal scientific platforms: PubMed, Scopus, Medline, Embase and Google scholar.

Using the MeSH database, initially, we found a total of 1326 articles matching “hypothermia, hypothermia AND emergency departments, emergency medicine, crowding, emergency medicine”. A second selection then reduced the number of suitable papers to 268, from which we then furthermore excluded meeting abstracts, books and unavailable manuscripts, and kept only relevant articles related to oncology and emergency medicine, critical or intensive care medicine and acute medicine. The reference lists of each article were reviewed to find further relevant articles to add. Eventually, 106 papers were analyzed in this clinical review.

## 3. Results

### 3.1. Definition

Hypothermia is defined as a reduction in core body temperature to less than 35.0 °C [13,14,15,16,17,18,19].

Hypothermia can be classified, according to its severity, as mild, moderate or severe [16,18,20]

-Mild hypothermia: 32 °C < core body temperature < 35 °C.-Moderate hypothermia: 28 °C < core body temperature < 32 °C.-Severe hypothermia: core body temperature < 28 °C.

When taking into consideration the etiopathology of hypothermia, it is important to differentiate between spontaneous and induced hypothermia.

Spontaneous hypothermia can, in turn, be divided into the following:-Primary hypothermia is due to environmental exposure, with no underlying medical condition causing the disruption of temperature regulation [21]. It therefore occurs when a person is exposed to the cold without adequate protection.-Secondary hypothermia is a complication of pathological and paraphysiological conditions that determine the hypothermia itself or cause either an alteration of the thermoregulation mechanisms, reduced heat production or increased heat dispersion.

Induced hypothermia can, in turn, be divided into:


-Therapeutic hypothermia;-Trauma-induced hypothermia [22].


Hypothermia is caused by a disturbance in an individual’s thermal homeostasis which causes physiological dysfunctions in several vital organs (Table 1).

Among the effects of hypothermia, it is important to remember a reduction in oxygenation of the tissues due to peripheral vasoconstriction, a reduction in myocardial contractility and a decreased ability to extract oxygen from hemoglobin [23,24,25,26]. Depending on the severity of the cold stress and the duration of exposure to it, there may be various adverse effects, which can be as severe as death [16,18,20]. Hence, the recommendation of some authors is to monitor the physiological state of the patient more than the temperature itself [19].

### 3.2. Physiopathology

Thermal homeostasis is the maintenance of a stable temperature between 34 °C and 37.5 °C due to automatic regulation mechanisms that allow the body to be a system open to the external environment in continuous thermal equilibrium. These mechanisms are regulated by a thermoregulation system that balances the production of heat and its dispersion (Figure 1) [27,28].

The process of thermoregulation takes place at the level of the preoptic nucleus of the hypothalamus, which acts by activating subsequent effectors. In turn, this center continuously receives information on the state of temperature from thermal receptors located in the skin, aorta, arteries, and brain. The organs mainly involved in the production of heat in basal conditions are the liver and heart, followed in dynamic conditions by the musculoskeletal system. On the other hand, the organs most involved in the dispersion of heat are the skin, which undertakes an estimated 90% of the dispersion, and the respiratory system, which is responsible for the remaining 10% [29,30,31].

The mechanisms (Figure 1) by which the skin and lungs can dissipate heat are [32]:Conduction: the transfer of heat to a cooler object through direct contact (for example, when immersed in cold water).Convection: the transfer of heat at the body surface via air circulation (for example, when exposed to cold air or wind).Evaporation: cooling of the skin surfaces when sweat changes from a liquid to a vapor form.Radiation: occurs through the transmission of electromagnetic waves.

Of these mechanisms, the ones most frequently responsible for accidental or spontaneous hypothermia are conduction and convection.

The automatic thermo-regulatory response to maintain hemostasis occurs through several mechanisms (Figure 1):-The redistribution of blood flow via vasoconstriction and a subsequent reduction in blood volume directed towards the skin and subcutis to reduce heat loss;-Shivering, which generates heat via muscle contraction;-Decreased sweating;-Increased thyroid activity due to hypothalamic stimulus;-Increased adrenal activity due to hypothalamic stimulus.

The stimulus of cold also determines, in a conscious subject who is able to perform them, some behavioral reactions (Figure 1):-Increased physical activity;-A shift to warmer environments;-Wearing protection against the cold;-Taking off wet clothes and replacing them with dry clothes.

Hypothermia therefore develops when there is a malfunction in one of the stages of thermal regulation or when there is either increased heat loss or reduced heat production that exceeds the compensation capacity of the regulatory centers.

### 3.3. Etiology and Risk Factors

Spontaneous hypothermia (primary) can occur due to unfavorable environmental conditions that exceed the body’s ability to defend itself from the cold.

Environmental causes related to hypothermia are, for example, exposure to the cold with wet clothes or without adequate protection or simply exposure to excessively cold temperatures.

Spontaneous hypothermia (secondary) can also occur due to internal causes, such as pathological states, which result in reduced heat production or increased heat loss or ineffectiveness of the thermal regulation system.

Internal causes that can lead to hypothermia are:-Impaired thermoregulation (Figure 2): This may be paraphysiological in the extreme ages of life (geriatric people or infants). Impaired thermoregulation may also occur following pathological conditions such as mental illnesses, or pathologies of the nervous system (such as Parkinson’s disease, stroke, multiple sclerosis, hypothalamic dysfunction, brain trauma or subarachnoid hemorrhage) or pathologies affecting the peripheral nervous system (neuropathies, diabetes mellitus, trauma to a section of the spinal cord). Impaired thermoregulation may also be due to an iatrogenic effect of drugs such as anxiolytics, antidepressants, phenothiazines, barbiturates, opioids, antipsychotics, oral antihyperglycemics, β-blockers and α-blockers [33,34].-Increased heat loss (Figure 2): This can be caused by dermatologic diseases such as burns, exfoliative dermatitis or psoriasis [35,36,37]. Heat loss can also be increased in the case of indulgent habits such as alcohol intake [38,39] but it can also have an iatrogenic origin, as in the case of cold infusions, the transfusion of cold hematopoietic components, hasty birth or anesthesia.-Decreased heat production (Figure 2): This is secondary to endocrine dysfunction (hypopituitarism, hypothyroidism, hypoadrenalism and hypoglycemia), malnutrition or either conditions or drugs that alter the level of consciousness, causing an impaired shivering mechanism.

Hypothermia can also be induced intentionally in medical settings for its neuroprotective effect following cardiac arrest, stroke or traumatic brain or spinal cord injury.

Risk factors that can predispose a patient to spontaneous hypothermia include external environmental factors not related to the individual, such as winter sports activities, immersion in cold water, wearing wet or insufficiently warm clothing or a lack of areas in which to shelter from cold [16,17].

There are also risk factors determined by physiological or paraphysiological conditions, such as the extreme ages of life. Infants and the elderly are fragile populations at high risk for developing hypothermia as they have less effective regulatory mechanisms; they are less able to generate heat through the shivering mechanism, both due to reduced muscle mass and, in the elderly, reduced function of their reflexes, and peripheral vasoconstriction is usually less effective for the retention of heat. In addition, it should be remembered that children are more susceptible to hypothermia than adults since they have a higher body surface index, causing a greater loss of heat via convection [40]. While it is true that children generate more metabolic heat than adults, enough to maintain body heat during exercise, this is not true during prolonged rest, during which they are more at risk of developing hypothermia [41]. In the elderly, the ability to self-regulate their internal temperature is also impaired due to possible pathologies affecting the cardiovascular, endocrinological and musculoskeletal systems. It should be remembered that this fragile category of patients are often polymedicated or have numerous comorbidities that can further compromise their ability to respond to cold stress.

The risk of hypothermia is also greater in malnourished patients or in those who abuse alcoholic drinks [38,39].

In addition to the aforementioned pathologies that can compromise thermogenic homeostasis, it is also important to remember patients affected by psychiatric disorders, septic/shocked patients or those whose mobility is limited by recent disabilities or injuries, who are therefore unable to implement normal behavioral mechanisms in response to the cold or who have an acutely impaired autonomic response capacity to reactions such as heat generation with muscle contraction [10,17,18,40,42].

The most common sources of temperature loss in trauma patients include exposure (environmental, as well as cavitary), the administration of i.v. fluids, anesthesia/loss of shivering mechanisms, and blood loss per se [17,22,43], but also possible brain injuries and burns. Hypothermia is one of the detrimental physiological effects that come with severe injury and hemorrhage (and thus, acidosis and coagulopathy). Awareness and the diligent screening of current body temperature in injured patients is necessary to detect, prevent and treat further temperature loss [44].

### 3.4. From Pathophysiology to Clinical Manifestations

Any situation that results in significant sudden heat loss can result in hypothermia because the mechanisms that cause heat build-up are slower [7] than the ones responsible for heat dispersion (60% of heat is lost through radiation, 10–15% through conduction and convection and 25–30% through evaporation and respiratory expiration [18]).

As already mentioned above, thermal homeostasis is performed by the regulatory center in the preoptic area of the hypothalamus and integrated by effectors of autonomous mechanisms and by behavioral attitudes. Resistance to hypothermia therefore improves both the correctness of the behavioral adaptation and the integrity of the complex regulatory center.

The main physiological responses to cold stress occur at the level of several systems and are hereby described in detail.

#### 3.4.1. Cardiovascular System

In cases of mild hyperthermia, the thermoregulatory center can determine increased secretion of catecholamines following the activation of the sympathetic nervous system. The result is an increase in mean arterial pressure and cardiac output by means of tachycardia and increased peripheral resistance [45]. On the other hand, when moderate hypothermia occurs, cardiovascular function gradually deteriorates [46,47].

Cardiac conduction is also affected by hypothermia through decreased pacemaker cell activity, resulting in bradycardia [48]. This bradycardia is typically refractory to atropine, because it is not vaguely mediated [49,50]. If cold stress causes severe hypothermia, there is a risk of developing atrial and ventricular arrhythmias due to a reduction in transmembrane resting potentials [50,51,52].

#### 3.4.2. Central and Peripheral Nervous Systems

Hypothermia causes a linear decrease in central nervous system metabolism as the internal body temperature decreases; more specifically, there is a decrease in oxygen consumption of about 6% for each reduction of 1 °C after the first loss [53,54,55,56]. This is reflected in a brain syndrome characterized by behavioral changes, amnesia, disorientation, dysarthria and ataxia [45]. Consciousness is progressively impaired with the progression of hypothermia.

#### 3.4.3. Respiratory System

Hypothermia can affect pulmonary functions via different mechanisms: pulmonary vascular resistance is increased with increased cutaneous vasoconstriction [57,58,59].

As the exposure to cold progresses and hypothermia becomes more severe, there is a progressive decrease in tidal volume and respiratory rate, a decrease in thoracic compliance and an increase in dead space.

In the case of severe hypothermia, the control of respiration is depressed until a picture of respiratory acidosis is formed, due to tissue accumulation of CO_2_ [19,40,45].

#### 3.4.4. Fluid Shifts, Electrolyte Balance and Kidney Function

Cold stress causes peripheral vasoconstriction with sequestration of the plasma volume and an increase in hematocrit [60]. This phenomenon, in conjunction with the loss of distal tubular water and the reabsorption of electrolytes, would determine the suppression of the release of the antidiuretic hormone, causing so-called “cold diuresis” [61].

Cold stress is also responsible for a reduction in renal oxygen consumption, compromising tubular function. In cases of severe hypoxia, the kidney is no longer able to guarantee serum levels of electrolytes, and a reduction in the tubular secretion of hydrogen ions is observed [62]. In these cases of impaired renal function, intense muscle contraction following shivering can lead to rhabdomyolysis.

#### 3.4.5. Blood and Coagulation Parameters

Hypothermia can be responsible for an increased risk of thrombosis through several mechanisms [63,64,65,66]. Blood viscosity increases due to the previously discussed hematological concentration and increase in hematocrit [45,67]. Hypothermia also causes an alteration in the homeostatic enzymatic activity of coagulation factors, leading to coagulation disorders [64,68,69,70]. An increase in spontaneous platelet activation is observed at temperatures below 37 degrees Celsius, and hypothermia also causes bleeding due to thrombocytopenia following hepatic sequestration [66].

With moderate hypothermia, a reduction in fibrinogen synthesis is also observed, with a consequent increased risk of bleeding [69,71].

#### 3.4.6. Endocrine and Metabolic Responses

The release of catecholamines stimulates thermogenesis when the core temperature is greater than 32 °C [72]. Catecholamine-induced glycogenolysis also induces hyperglycemia, with a decrease in insulin release, the inhibition of insulin transport (that becomes inactive at core temperatures < 31 °C), a decrease in liver enzyme function, and a decrease in the renal clearance of glucose that promotes hyperglycemia. However, glycemic levels during hypothermia have been reported to be both high and low [73], thus indicating that glycemic control is primarily dependent on the metabolic state of the patient [56,57]. Interestingly, pancreatitis is a common finding in autopsies of hypothermic patients [74].

On a side note, acute cold exposure causes an increase in metabolic heat production [75].

#### 3.4.7. Gastrointestinal Tract

Intestinal motility decreases below about 34.8 °C, resulting in an ileus when the temperature falls below 28.8 °C; therefore, a nasogastric tube should be placed to reduce the chance of aspiration in hypothermic patients.

The absorption of medication given orally or via a nasogastric tube will also be impaired in this situation, and this administration route should therefore be avoided. Punctate hemorrhages may occur throughout the gastrointestinal tract. Hepatic impairment can develop, probably as a consequence of reduced cardiac output, and the decreased metabolic clearance of lactic acid contributes to acidosis. Pancreatitis frequently occurs as a consequence of hypothermia, being found at autopsy in 20–30% of cases, and mildly elevated serum amylase without clinical evidence of pancreatitis is even more common, being present in 50% of patients in one series [76].

### 3.5. Clinical Diagnosis

The diagnosis of accidental hypothermia includes (i) a history or evidence of exposure to cold stress and (ii) an internal temperature < 35 °C.

For a correct diagnosis, it is necessary to use the correct instrument (thermometer). A low-reading thermometer (capable of measuring temperatures up to 25 °C) is preferred, and it should be used via the rectal, esophageal (more suitable) or bladder route. Bladder and rectal temperatures should not be used in critically ill patients during rewarming [77,78].

Tympanic temperature measurement may also represent a practical, non-invasive approach to core temperature monitoring in an emergency setting [79]. It has indeed been demonstrated that tympanic temperature is a good index of core temperature and it accurately reflects both esophageal and bladder temperatures with a very small discrepancy [80,81].

The signs and symptoms that can guide us to a diagnosis depend on the severity of hypothermia. The various physiological alterations that occur in hypothermia can be grouped summarily according to the degree of severity of the hypothermia itself [82].

In mild hypothermia, the patient is conscious and presents vigorous shivering, increased cardiac output due to increased peripheral resistance, and tachycardia; they also present tachypnoea, and in cases of the persistence of the cold stress, neurological signs such as dysarthria, ataxia and motor impediment are observed. Cold diuresis occurs secondary to peripheral vasoconstriction, which is also responsible for cold extremities and pallor. At the gastrointestinal level, cold stress can lead to the formation of gastric ulcers and pancreatitis. For what concerns the blood system, the risk of thrombosis due to hemoconcentration and the risk of bleeding due to the inactivation of coagulation factors are already seen in the early stages of hypothermia.

Moderate hypothermia is characterized by decreased cardiac output and blood pressure, hypoventilation and hyporeflexia. The loss of the shivering mechanism is also observed. The resulting picture ranges from an impairment of mental function up to a loss of consciousness. The gross impairment of motor control, cessation of shivering, cyanosis, muscle rigidity, mydriasis, atrial or ventricular cardiac dysrhythmias and bradycardia also occur. In this phase, the behavioral defenses are compromised in some subjects, who paradoxically undress.

If the cold stress perdures, or if the regulatory mechanisms are so compromised as to progress to a state of severe hypothermia, the patient may present in a state of shock or pre-shock with hypotension, pulmonary congestion, edema, muscle rigidity, areflexia, oliguria and coma.

Tissues have decreased oxygen consumption at lower temperatures; at 28 °C, oxygen consumption is reduced by about 50%, and at 22 °C by about 75%.

More severe pictures include spontaneous ventricular fibrillation and cardiac arrest. [9] Severe cases can mimic death. At 18 °C, the brain can tolerate ten times longer periods of cardiac arrest than at 37 °C. When faced with a patient in cardiac arrest who is hypothermic, it is important to remember that they should not be declared dead until they have been rewarmed [10,14,15,16,40].

When the core temperature cannot readily be measured, the Swiss staging system, which distinguishes among five levels of hypothermia based on clinical appearance, can be a useful tool to determine the severity of hypothermia:-HT I: clear consciousness and shivering.-HT II: impaired consciousness without shivering.-HT III: unconsciousness.-HT IV: apparent death.-HT V: death due to irreversible hypothermia [20,40].

### 3.6. Laboratory Studies

After making the diagnosis of hypothermia, laboratory evaluation should be undertaken to identify potential complications and comorbidities, including lactic acidosis, rhabdomyolysis, bleeding diathesis, infection, renal failure and pancreatitis. Standard laboratory evaluation should include finger-stick glucose, complete blood count, coagulation tests and a metabolic panel including kidney (creatinine and electrolytes), liver and pancreatic function tests.

Due to the effects of cold stress on coagulation enzymes, prolongation of prothrombin time and of tests of activated partial thromboplastin time should also be performed [83].

Arterial blood gases are difficult to interpret in people with hypothermia. The pH of neutrality increases upon cooling. Blood gas analyzers heat samples to 37 °C before analysis and may calculate corrected values according to the core temperature of the patient. However, some authorities currently support the use of uncorrected arterial blood gas measurements to guide treatment. Metabolic acidosis is a feature of severe hypothermia and may be a consequence of lactic acid generation due to poor tissue perfusion. Type I or type II respiratory failure may also be identified when interpreting the results of the arterial blood gas analysis.

Due to the different arrhythmias that are often present in hypothermic patients, an ECG is necessary. The ECG may show an enlargement of the QRS complex or a J wave (Osborn’s wave) plus a prolonged QT interval and T wave inversion [84,85]. These signs are not pathognomonic, but are needed both to suspect and to understand the severity of the cold stress to which the organism has been subjected [12,86]. In advanced stages, a reduction in cardiac output may also occur due to both an increase in afterload and a decrease in heart rate [15].

Imaging should be dictated by clinical scenarios; a chest X-ray is not uncommon for severely hypothermic patients in order to detect signs of pulmonary edema [87,88].

An EEG could be necessary based on clinical circumstances. An electroencephalogram (EEG) is abnormal at a core temperature < 33.5 °C, suppressed at 22 °C, and becomes silent between 19 and 20 °C [54,89].

## 4. Discussion and Contextualization

### 4.1. Point of View of the Emergency Departments: General Aspects and Management

#### 4.1.1. Management of Hypothermic Patients in the Emergency Department (ED)

During the coldest seasons, or in colder areas such as the mountains, patients with hypothermia can present to the ED.

It has already been demonstrated that it is often difficult to collect a proper anamnesis during triage, especially in crowded emergency departments in which the outcomes have already been demonstrated to be worse per se [90,91,92,93,94,95]. In these circumstances, it is necessary to pay close attention to the patient in order to highlight the risk factors for worse outcomes, especially those related to pre-existing pathologies.

The risk factors for developing hypothermia, as well as heat diseases, include being part of a fragile population, such as the elderly, who are already a population at risk for worse outcomes in the emergency department for many diseases [90,96,97,98], and who often present with pathologies that can be underestimated at triage [99,100].

Besides the difficulty of finding a correct medical history, it is also important to consider the fact that the doctor of the emergency department does not have proof of the severity of the prehospital conditions of the patient when the patient arrives in a territorial system.

Important information can therefore be lost, and when considering a state of shock or pre-shock, the need to exclude many differential diagnoses can slow down the correct management and treatment.

Older people may have a reduced tolerance to low temperatures due to loss of muscle mass, as well as cognitive deficits that might determine behavioral alterations, and therefore, correct dressing in cold environments.

In this case, older patients may develop more severe frameworks than one might expect from cold exposure alone.

It is also important to keep in mind that the elderly manifest clinical pictures that can be insidious, and they may not report symptoms correctly, either due to language limitations or due to underestimation of the symptom (the elderly often live with symptoms such as pain and unease). Older patients may also experience delirium, which, if not recognized and treated promptly, can lead to poor outcomes.

In patients belonging to the most fragile categories, and in the most compromised ones, it is useful to perform blood tests, including blood counts, biochemistry, liver enzymes, electrolytes, renal function and creatine phosphokinase (CPK) dosage.

An ultrasound evaluation will allow for an estimation of the ventricular filling pressure and the patient’s volume status [101,102].

Patients with severe hypothermia or major clinical pictures should then be treated and stabilized in observation units, which have already proven effective for large categories of patients [98,103,104].

Providing plans for any hyper-influxes in cold seasons can be important, especially if in conjunction with major sporting or entertainment events, as already demonstrated in other fields [4].

#### 4.1.2. Inpatient Treatment of Moderate-to-Severe Hypothermia

Once hypothermia is recognized, prompt treatment is necessary to avert preventable death.

Wet or damp clothing should be removed without delay, and the individual should be removed from wind or rain and moved to a warm, dry, sheltered environment. Rewarming should begin as soon as possible. The treatment of a patient with severe hypothermia should be focused on restoring body temperature and supporting vital functions.

In case of hypothermia, it will be necessary to administer humidified and heated oxygen. Moreover, all patients with hypothermia should be assumed to be dehydrated and receive supplemental fluid resuscitation, preferably with rewarmed 5% glucose [16].

Rewarming techniques are divided into passive external rewarming, active external rewarming, and active internal core rewarming. The degree of hypothermia determines the techniques implemented [3,40].

Passive external rewarming (Table 2) is centered on protecting the patient from further heat loss whilst thermoregulation helps raise the body temperature. Passive external rewarming is therefore used for mild hypothermia, in which thermoregulation mechanisms are still functional. After wet clothing is removed, the patient is covered with blankets or other types of insulation. The resulting reduction in heat loss combines with the patient’s intrinsic heat production to produce rewarming. Passive external rewarming requires sufficient physiologic reserves to generate heat by shivering or by increasing the metabolic rate.

Active external rewarming (Table 2) relies on the delivery of heat to the surface of the body. During active external rewarming, some combination of warm blankets, heating pads, radiant heat, warm baths or forced warm air is applied directly to the patient’s skin. Friction massage should be avoided as this can exacerbate tissue damage from frostbite [15,105]. These methods are indicated for moderate-to-severe hypothermia and for patients with mild hypothermia who are unstable, lack physiologic reserve, or fail to respond to passive external rewarming. Especially in chronic hypothermia with dehydration, rewarming of the trunk should be undertaken before the extremities, in order to minimize core temperature afterdrop with associated hypotension and acidemia due to arterial vasodilation [45].

Active internal (core) rewarming (Table 2) relies on the delivery of heat to the interior of the body. Active internal rewarming (also called active core rewarming) is the most aggressive strategy. It can be used alone or combined with active external rewarming in patients with severe hypothermia (<28 °C) or patients with moderate hypothermia who fail to respond to less aggressive measures. In addition to IV administration of warmed crystalloid (40 to 42 °C), effective techniques include warmed humidified oxygen, irrigation of the peritoneum or the thorax (via the pleural space) with warmed isotonic crystalloid and extracorporeal blood rewarming. Extracorporeal blood rewarming is performed in extreme cases (e.g., cardiac arrest, frozen limbs) or when rewarming is inadequate despite the other measures described here. Several techniques can be used to treat hypothermic patients by rewarming blood outside the body: venovenous rewarming, hemodialysis, continuous arteriovenous rewarming (CAVR), cardiopulmonary bypass (CPB) and extracorporeal membrane oxygenation (ECMO) [13].

#### 4.1.3. CPR (Cardiopulmonary Resuscitation) and Management of Arrhythmias

It is important to note that CPR in hypothermic patients is particularly tiring due to the decreased elasticity of the rib cage. For temperatures below 30 °C, defibrillation may be ineffective [106,107]. It is therefore necessary to start heating maneuvers and carry out subsequent defibrillation after an increase of 1–2 °C or after the temperature has surpassed 30 °C. It should be remembered that arrhythmias often disappear with warming. The use of vasoactive drugs is controversial and not routinely recommended, especially at temperatures below 30 °C [108,109]. Once 30 °C has been reached, the ERC 2020 recommendations [110] advise lengthening the intervals at which to administer adrenaline to 6–10 min. Resuscitation maneuvers should continue until the patient has been rewarmed [111].

Vital signs should be closely monitored while performing the measures to treat hypothermia; peripheral pulses may be difficult to palpate in hypothermic patients.

Any patient with suspected hypothermia who has signs of cardiac dysrhythmia should be moved gently to avoid precipitating paroxysmal ventricular fibrillation, as the myocardium is more sensitive to mechanical stimulation during deep hypothermia [15]. In situations where the EKG shows an organized cardiac electrical rhythm (other than fibrillation), cardiopulmonary resuscitation with chest compression should not be performed, despite the absence of a palpable pulse, because chest compressions may convert an adequate perfusing rhythm to ventricular fibrillation [17], unless ETCO2 monitoring is available, which confirms the absence of perfusion, or it is possible to perform an echocardiography, which documents the absence of effective cardiac activity in correspondence with the electrical activity [78].

Due to the aforementioned physiological and metabolic changes, particularly in the central nervous system, in cases of hypothermia, prolonged cardiac arrests do not necessarily lead to neurological damage, and cases of complete neurological recovery have been described in the literature even after long periods of arrest or after prolonged CPR [78,112,113,114,115,116,117].

In cases of cardiac arrest, CPR with compressions and ventilations should be carried out as in normothermic patients. If available, mechanical CPR devices should be considered in cases of prolonged rescue operations. In severely hypothermic patients in cardiac arrest, if continuous or mechanical CPR is not possible, intermittent CPR should be used [118].

The risk of ventricular fibrillation in a severely hypothermic patient increases as core temperature rises above 28 °C. Below that temperature, cardiac dysrhythmias tend to be refractory, so resuscitative efforts should continue until the absence of electrographic cardiac activity is documented after the core temperature has risen to 28–30 °C [119].

#### 4.1.4. Secondary Hypothermia in Acute Diseases

For hypothermic patients in whom attempts at rewarming fail, it is important to think about underlying diseases that could cause refractory hypothermia [33,34,120]. This is the case for hypothyroidism, hypoglycemia or iatrogenic hypothermia. These conditions are more frequently found among older polymorbid patients with severe clinical pictures. In this type of hypothermia, cold stress plays a role mainly due to the malfunctioning of the central regulatory systems. In these cases, in addition to heating, it is therefore also necessary to treat the concomitant pathologies: administer insulin for hyperglycemia in diabetes, glucose for hypoglycemia, cortisone for acute adrenal insufficiency or broad-spectrum antibiotic therapy for sepsis.

#### 4.1.5. Recommendations for Future Research

Hypothermia is a condition that is most frequently found and managed in prehospital and primary care settings.

Previous studies are mainly focused on the management of hypothermic patients in the a prehospital setting. However, few studies are focused on the interdisciplinary management of these patients in emergency departments.

Future research should therefore focus on the optimization of the management of hypothermia in the emergency department, which remains one of the main challenges of modern emergency medicine.

### 4.2. Risk Management and the Forensic Point of View

By definition, the diagnosis of hypothermia is based on the measurement of core body temperature, which must be below 35 °C (95.0 °F) [19,121]. Body temperature should preferably be measured at two different body sites.

The post-mortem diagnosis of hypothermia can rarely provide certainty, as many of the findings are nonspecific. In fact, to be comprehensive, the diagnosis of hypothermia must consider several factors, including clinical history and circumstantial data. Furthermore, other causes of death must be excluded [122].

#### 4.2.1. Post-Mortem Changes

In regard to post-mortem changes, cadaveric cooling (algor mortis) is clearly immediate, cadaveric rigidity (rigor mortis) is often confused with the crystallization of organic fluids, hypostases (livor mortis) are poorly represented and putrefactive processes slow down [123].

#### 4.2.2. External Findings

External examination of the body does not allow for the detection of common pathognomonic signs of hypothermia.

External findings in patients who have succumbed to hypothermia include red-brownish patches [124,125] usually appearing in correspondence with the extensor surfaces and large joints (elbows and knees), and less often on the face (especially the cheeks, chin and nose). In association with these typical patches, a cyanotic appearance [124,125] of the extremities, which may possibly also assume a whitish color (fingers and toes), may be observed. Lastly, systemic hypothermia implies the development of cold erythema (pernions) that can be detected upon external examination.

#### 4.2.3. Autopsy Findings

Among the internal findings in patients who have died with hypothermia, the presence of blackish erosions with a hemorrhagic background in the gastric mucosa, known as Wischnewski spots, have been described [126,127,128]. Acute pancreatitis [74,76] can be observed, as well, although it has been observed in less than half of all hypothermia deaths. At the dissection table, the expert forensic physician will have to scout the pancreas section for the presence of areas of fat necrosis [124] presenting a yellowish color, sometimes extended to involve the omentum and mesentery. In addition, aspects of perivascular microhemorrhages [124,125] have been described in the cerebral parenchyma, especially in the third ventricle. These, however, do not represent a specific finding.

#### 4.2.4. Microscopic Findings

Histologically, good conservation of tissues is observed in hypothermic patients, as autolytic processes are slowed by low temperatures.

In particular, cold erythema [123,124] shows intense hyperemia in the subepidermal soft tissue and mild dermal edema. Hemoglobin positivity is found immunohistochemically.

Furthermore, in regard to the Wischnewski spots [123], these are recognizable histologically due to their coloration, as well as immunohistochemically, thanks to hemoglobin positivity.

As a result of the high level of cold agglutinin and hemoconcentration, erythrocyte sludge may additionally be found in organs and tissues; this can lead to micro-infarcts found in organs such as the intestine or portal vein.

The presence of a fatty degeneration [61,124] of the renal tubular epithelial cells, cardiomyocytes and hepatocytes is also observed [128,129]. In the context of the anterior pituitary gland, the presence of cell vacuolization is found. Immunohistochemical findings [124] detected the presence of HSP70 in the kidney and, in other cases, the expression of ubiquitin. Lastly, the pancreas presents areas of microhemorrhage, fat necrosis [74,76,124] associated with leukocyte infiltration and, occasionally, intracytoplasmic vacuolization.

## Figures and Tables

**Figure 1 jpm-13-01690-f001:**
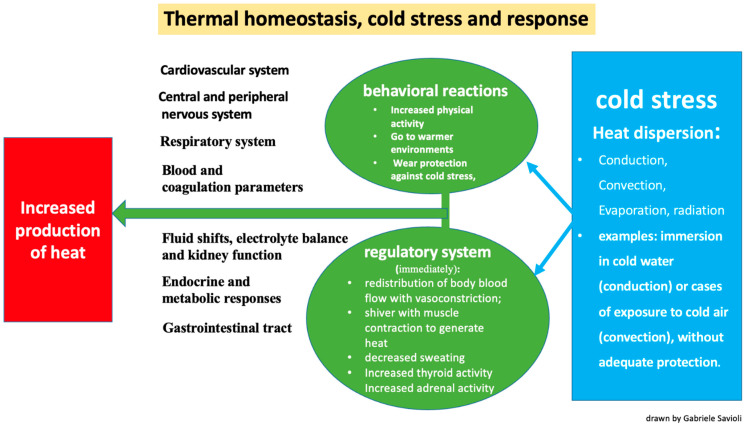
Thermal homeostasis and physiological response to cold stress.

**Figure 2 jpm-13-01690-f002:**
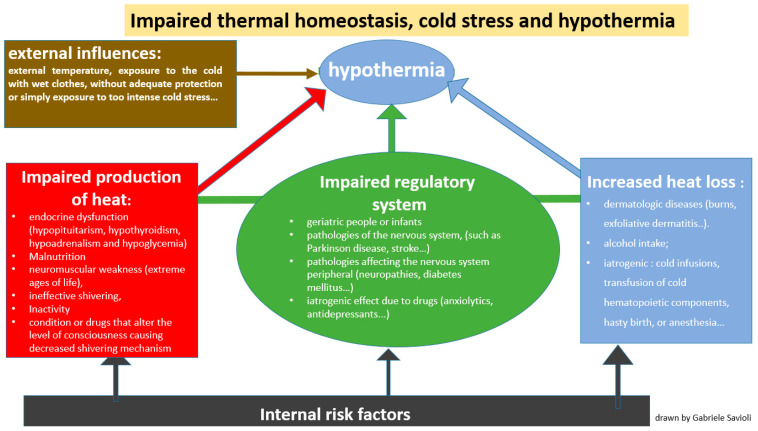
Mechanisms by which hypothermia develops: disruption of thermal homeostasis and dysregulated response to cold stress.

**Table 1 jpm-13-01690-t001:** Classification of hypothermia from etiopathology.

Classification from Etiopathology		
Spontaneous hypothermia	Primary hypothermia	Secondary hypothermia
Induced hypothermia	Therapeutic hypothermia	Trauma-induced hypothermia

**Table 2 jpm-13-01690-t002:** Inpatient treatment of moderate-to-severe hypothermia.

	Type of Rewarming		
	Passive External	Active External	Active Internal (Core)
When to adopt	Mild hypothermia, in which thermoregulation mechanisms are still functional.	Can be used for moderate-to-severe hypothermia and for patients with mild hypothermia who are unstable, lack physiologic reserve, or fail to respond to passive external rewarming	It can be used alone or combined with active external rewarming in patients with severe hypothermia (<28 °C) or patients with moderate hypothermia who fail to respond to less aggressive measures
What to do	After wet clothing is removed, the patient is covered with blankets or other types of insulation.	Relies on the delivery of heat to the surface of the body (some combination of warm blankets, heating pads, radiant heat, warm baths or forced warm air, is applied directly to the patient’s skin).	IV administration of warmed crystalloid (40 to 42 °C) or extracorporeal blood rewarming

## Data Availability

Not applicable.

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
