# Peer review of "Hypothermia: Beyond the Narrative Review—The Point of View of Emergency Physicians and Medico-Legal Considerations"

_jpm, 2023, doi:10.3390/jpm13121690_

Round 1
Reviewer 1 Report (Previous Reviewer 2)
Comments and Suggestions for Authors
Review
The main purpose of the article, as the title indicates, was to present recommendations regarding the diagnosis and management of a hypothermic patient in the emergency department and to present the medical and legal aspects of such admissions.
(1) The presented manuscript has the characteristics of a review article. However, the manuscript was submitted as an article and not as a review article. Authors themselves mention in abstract (line 26) and Methods (line 56) that the manuscript is a review article. I recommend to change the type of publication into review.
(2) The aim of the manuscript which is presented in the abstract should be repeated at the end of introduction.
(3) According to the author’s guidelines of Journal of Personalized Medicine, review should identify current gaps or problems in the analyzed issue and provide recommendations for future research. Above should be included in the separate section of discussion or in conclusions.
(4) There are some inappropriate self-citations that are not relevant to the data presented and should not be included. (80, 83, 84, 85, 88, 89). I advise to refrain from using not relevant self-citations. It rises ethical doubts.
(5) Considering that the article is a review, and that it mainly repeats previously published data, the number of co-authors, which is as many as twelve, raises great ethical doubts.
(6) There are some additional minor mistakes to improve:
a. Line 42 – I suggest to change “its” into mortality
b. Line 53 – I suggest to add “other” before European. UK is also in Europe.
c. Line 97 – please provide reference to “…. decreased ability to extract oxygen from hemoglobin.”
d. Line 132 – Sweating alone is responsible for dissipating an increase of heat not preserving heat. Therefore or include “decreased” before sweating or delete sweating form the list
e. Graph 1 in regulatory system - sweating should not be included or “decreased” should be added before sweating
f. Line 175 – add references
g. Lines 264-270 Please add information that hypothermia might increase a risk of thrombosis.
h. Line 298-300 – add information about potential use of tympanic measurements of core temperature in hypothermia
i. Line 333 – change à into a
j. Line 445 – change increased into decreased
k. Line 450 – provide reference
l. Lines 465-482 – changes references into numbers
m. Lines 493-496 – text is in Italian – it is appropriate but should be changes into English
Considering the above, in my opinion the manuscript is publishable but needs major improvements.
Comments on the Quality of English LanguageThe english quality is very good.
Author Response
(1) The presented manuscript could indeed be published as a review article.
(2) The aim of the manuscript which is presented in the abstract is now repeated at the end of introduction (highlighted).
(3) Recommendations for future research have been provided in a separate section of the discussion (highlighted).
(4) The considered inappropriate self-citations have been removed and modified.
(5) The authorships have been modified through the authorship change form.
(6) The additional minor mistakes have been corrected according to the suggestions proposed.
a. Line 42 – ok (highlighted)
b. Line 53 – ok (highlighted)
c. Line 97 – reference to “…. decreased ability to extract oxygen from hemoglobin” has been provided (highlighted)
d. Line 132 – “decreased” has been added before sweating (highlighted)
e. Graph 1 in regulatory system - “decreased” has been added before sweating
f. Line 175 – references have been added (highlighted)
g. Lines 264-270 - information that hypothermia might increase a risk of thrombosis has been added (highlighted)
h. Line 298-300 – information about potential use of tympanic measurements of core temperature in hypothermia has been added (highlighted)
i. Line 450 – references provided (highlighted)
l. Lines 465-482 – references have been rearranged into numbers (highlighted)
m. Lines 493-496 – text has been translated into English (highlighted)

Reviewer 2 Report (New Reviewer)
Comments and Suggestions for Authors
This is a well-written, good structured narrative review article. The figures are clear and easy to understand. However, I read nothing new from this article. All the provided information are the same with previous articles. It contributes little to clinical management. I suggest the authors find out real arguments about "hypothermia" and search for new evidence about these arguments. Another option is focused on an uncertained issue about "hypothermia" and tried to resolve it.
Author Response
Dear reviewer, we thank you for your comment which allows us to further enhance our work. The most innovative part of our article lies in fact in the point of view of the emergency doctor and the forensic doctor. We have therefore underlined this in the title and expanded the space dedicated to them.
(highlighted in blue)

Round 2
Reviewer 1 Report (Previous Reviewer 2)
Comments and Suggestions for Authors
I would like to thank the Authors for implementing the corrections I suggested in the article. I would only suggest two changes:
- change the type of publication onto review
- change in Graph 1 – please add “heat” after ‘to generate’ and add separate subpoint for ‘decreased sweating’ to make it more clear.
Kind regards
Author Response
- the type of publication has been already changed onto review
- the change in Graph 1 has been made
We thank you for your suggestions that helped improve the quality of the article.
Reviewer 2 Report (New Reviewer)
Comments and Suggestions for Authors
The authors changed the title of manuscript and focused more on the point of view of emergency physician. Additionally, the authors added more comments on geriatric population in the Discussion section. I think these changes improve the value of this manuscript.
Though the authors did not offer new scientific evidence in "Hypothermia" issue. The abundance of references provide sufficient details from the etiology to managements of patients with hypothermia.
Author Response
We thank you for your comments that helped improve the quality of the manuscript.
This manuscript is a resubmission of an earlier submission. The following is a list of the peer review reports and author responses from that submission.
Round 1
Reviewer 1 Report
Comments and Suggestions for Authors
Hypothermia is a widespread condition all over the world, with a high risk of mortality in the pre-hospital and in-hospital setting, which has been a widespread conundrum, so this is a good topic and it has been 5 years since the latest special subject about hypothermia (DOI: 10.1001/jama.2018.0749). This paper has described the main specificities of the diagnosis and treatment of hypothermia through the consideration of the physiological changes that occur in the hypothermic patient, which is a comprehensive and systematic exposition, however, there is still some space to improve for a better read.
(1) Add some tables to conclude and better exhibition, such as tables in “1. Definition”, “6. Clinical to Diagnosis”, “7. Laboratory Studies”, “8. Point of View of the Emergency Department: General Aspect and Management”, and “9. Risk Management and the Forensic Point of View;
(2)Optimize diagrams in “3. Physiopathology” and “4. Etiology and Risk Factors”;
Reviewer 2 Report
Comments and Suggestions for Authors
Review
The presented manuscript has the characteristics of a review article. However, the manuscript was submitted as an article and not as a review article. The main purpose of the article, as the title indicates, was to present recommendations regarding the diagnosis and management of a hypothermic patient in the emergency department and to present the medical and legal aspects of such admissions.
Recommendations for the diagnosis and management of hypothermic patients in pre-hospital and emergency departments are published regularly by the European Resuscitation Council (ERC) (latest version in 2021) and the Wilderness Medical Society (WMS) (latest version in 2019). The ERC and WMS guidelines are complete, so writing another article on the same topic seems unnecessary and not interesting to the reader. In addition, an article repeating previously published recommendations does not bring anything new to science. The aim to analyze the medical and legal aspects of admissions of hypothermic patients is interesting.
The article consists of 9 parts and is structured more like a textbook chapter than a scientific article. The purpose of the study is not clearly presented. The methodology is not described – both the original and the review article must present the methodology. There is no discussions and there are no conclusions.
The presented recommendations regarding the diagnosis and treatment of patients with hypothermia are very general compared to the ERC and WMS guidelines. Some of the information is misleading (for example in line 413 - authors do not recommend CPR for hypothermic patients with an organized electrical rhythm in the absence of a palpable pulse). The ERC and WMS guidelines recommend in such cases to confirm cardiac arrest with ETCO2 and ultrasound, and after its confirmation, start cardiopulmonary resuscitation. The authors do not mention this fact.
It is a pity that the medico-legal part of the article is also superficial and deals mainly with the forensic aspects of the hypothermic patient.
There are many simple spelling errors (lines 133, 137, 274, 325, 413). The format of the article is not uniform, the size and type of font change.
The article contains 62 references, of which 13 are self-citations by the first author. There are 11 inappropriate self-citations not related to the management of hypothermic patients and related to (Clostridium tetani infection in the elderly; impact of the Covid-19 pandemic on emergency department overcrowding; injuries in elderly patients; minor head injuries; acute heart failure in elderly patients age; cardiac patients; echocardiography; acute heart failure; acute pulmonary embolism).
Considering that the article is neither research nor review, and that it mainly repeats previously published data, the number of co-authors, which is as many as thirteen, raises great ethical doubts.
Considering the above, in my opinion the manuscript should be rejected from publication.
Reviewer 3 Report
Comments and Suggestions for Authors
Abbreviations must be deciphered at the first mention.
………. (CPR) Cardiopulmonary Resuscitation – or Computerized Patient Record?
………. (CNS) Central Nervous System – or – Community Network Services?
………. (ED) Emergency Department – or – Erectile Dysfunction?
………. (CPK) dosage, etc.